# Cyclin-Dependent Kinase Subunit 2 (CKS2) as a Prognostic Marker for Stages I–III Invasive Non-Mucinous Lung Adenocarcinoma and Its Role in Affecting Drug Sensitivity

**DOI:** 10.3390/cells11162611

**Published:** 2022-08-22

**Authors:** Junkai Feng, Menglong Hu, Zongkuo Li, Guiming Hu, Yuting Han, Yan Zhang, Min Zhang, Jingli Ren

**Affiliations:** 1Department of Pathology, The Second Affiliated Hospital of Zhengzhou University, Zhengzhou 450014, China; 2Academy of Medical Sciences, Zhengzhou University, Zhengzhou 450001, China

**Keywords:** CKS2, invasive non-mucinous adenocarcinoma, histological subtype, prognosis, carboplatin, paclitaxel

## Abstract

With the aim of improving the prognosis of patients with lung adenocarcinoma (LUAD), we identified the biomarker related to the sensitivity of patients to chemotherapy drugs and explored the potential mechanisms. As a cell cycle-related protein, CKS2 has an essential role to play in tumor progression and prognosis. CKS2 expression was measured using TCGA RNA-sequencing data and immunohistochemistry. The sensitivity data of tumor cells to chemotherapeutic drugs for lung cancer was acquired from the Cancer Therapeutics Response Portal (CTRP) database. A range of bioinformatics methods was used to explore the mechanisms of CKS2 upregulation. The biological functions of CKS2 were predicted using GO and KEGG enrichment analysis, as well as GSEA. CKS2 expression was up-regulated in stages I–III invasive non-mucinous lung adenocarcinoma and varied significantly between various histological subtypes. High CKS2 expression worsened the prognosis of patients. The CKS2 expression level was linked to the sensitivity of LUAD cells to carboplatin and paclitaxel. CKS2 upregulation was associated with the immune microenvironment, mRNA methylation, and competing endogenous RNAs (ceRNAs). CKS2 can serve as a diagnostic and prognostic biomarker for stages I–III invasive non-mucinous lung adenocarcinoma and modulate the effect of paclitaxel and carboplatin by regulating microtubule binding and influencing carboplatin binding to DNA.

## 1. Introduction

Lung cancer remains one of the malignant tumors with the highest incidence [1]. Non-small cell lung cancer (NSCLC) is the most frequent histological classification for lung cancer [2], with LUAD accounting for roughly 40% to 50% of the cases [3]. LUAD includes invasive non-mucinous adenocarcinoma, invasive mucinous adenocarcinoma, minimally invasive adenocarcinoma (MIA), intestinal-type adenocarcinoma, fetal adenocarcinoma, and colloid adenocarcinoma [4], of which the most common is invasive non-mucinous adenocarcinoma. The prominent histological subtypes of invasive non-mucinous adenocarcinoma are Lepidic (LEP), Acinar (ACI), Papillary (PAP), Micropapillary (MIP), and Solid (SOL) subtypes [4,5]. The prognosis of different histological subtypes varies, with the lepidic subtype having the best prognosis and both solid and micropapillary subtypes having the worst [6,7]. In addition, the Cribriform (CRI) structure is a special subtype with a poor prognosis. The majority of NSCLC patients have progressed to an advanced stage at diagnosis, in which the patients have usually developed lymph node metastasis or even distant organ metastasis, and thereby their prognosis is dismal [8]. Despite recent advances in medical technology, there is still a lack of reliable indicators with clinical significance for LUAD patients. Therefore, to enhance the prognosis of LUAD patients, finding an effective prognostic molecular marker is essential.

Cyclin-dependent kinase subunit 2 (CKS2), located on human chromosome 9q22 [9], belongs to the cyclin-dependent kinase (CDK) binding protein family. CKS2 plays a crucial function by binding to the catalytic subunit of cyclin-dependent kinases [10], and silencing it can arrest the cell cycle in the G2 phase [11]. In recent years, CKS2 has been reported to be strongly expressed in multiple malignant tumors, including gastric cancer [12], liver cancer [13], bladder cancer [14], and breast cancer [15]. CKS2 overexpression accelerated tumor progression and worsened the prognosis for LUAD patients. Further, immunohistochemistry revealed that CKS2 expression differed between various histological subtypes of invasive non-mucinous lung adenocarcinoma. However, some histological subtypes with high-grade structures were associated with poor prognosis [6,16], indicating that the prognostic significance of CKS2 in invasive non-mucinous lung adenocarcinoma deserves further research.

We have identified a prognosis-related molecule, CKS2, which is not only significantly up-regulated in invasive non-mucinous lung adenocarcinoma but also expressed differentially in various histological subtypes. CKS2 expression exhibited an inverse correlation with drug sensitivity to carboplatin and a positive correlation with drug sensitivity to paclitaxel. The expression level of CKS2 was regulated by methylation levels and the immune microenvironment, as well as by ceRNAs. This research reveals that CKS2 is a biomarker correlated with poor prognosis in invasive non-mucinous lung adenocarcinoma and, more importantly, that it is linked to sensitivity to various chemotherapeutic agents in LUAD patients, which may provide further scientific insights into the clinically individualized treatment.

## 2. Materials and Methods

### 2.1. Clinicopathological Slides

We collected the clinicopathological slides of 90 stages I–III patients with invasive non-mucinous lung adenocarcinoma who had been treated with radical resection without neoadjuvant chemotherapy from the Second Affiliated Hospital of Zhengzhou University. In addition, the clinicopathological slides of 211 patients were obtained from the TCGA database (https://portal.gdc.cancer.gov/repository, accessed on 20 December 2021). All of the samples met the following inclusion criteria: (1) patients at stages I–III based on the eighth edition of AJCC; (2) pathologically confirmed primary invasive non-mucinous lung adenocarcinoma; (3) complete clinical information; (4) no other malignant tumors; (5) the histological subtypes and STAS status as diagnosed by two senior pathologists were consistent. This study was approved by the Ethics Committee of the Second Affiliated Hospital of Zhengzhou University (2021312). 

### 2.2. Transcriptome Data

The gene expression matrixes of 594 LUAD samples from TCGA (row counts, FPKM) were obtained, of which 59 were normal lung samples, and 535 were tumor tissue samples. After removing the samples without staging information, 472 patients at stages I–III were retained. The gene expression profile of 288 normal lung tissues was acquired from Genotype-Tissue Expression (GTEx, https://gtexportal.org/home/, accessed on 20 December 2021).

### 2.3. Immunohistochemical Staining

After the slides were repaired, they were treated with a 3% hydrogen peroxide solution for 10 min. The slides were then incubated at 4 °C overnight with diluted anti-CKS2 antibody (1:50, Abcam, EPR7946(2), Cambridge, UK) and then incubated with the secondary antibody for 30 min. Diaminobenzidine tetrachloride solution was added for color development.

The positive cell percentage score was defined as follow: 0, less than 5%; 1, 6% to 25%; 2, 26% to 50%; 3, 51% to 75%; 4, more than 75%. The following was used to assign the staining intensity score: 0, negative; 1, weak; 2, moderate; and 3, strong. The immunohistochemical scores were calculated by multiplying these two scores.

### 2.4. Acquiring Drug Sensitivity Data 

The CTRP database contains data on the sensitivity of different tumor cells to different chemotherapeutic drugs. We calculated the sensitivity of each LUAD tumor patient to different chemotherapeutic drugs through the ‘oncoprdict’ R package [17] based on the gene expression data (log2(TPM + 1)). The sensitivity data of three chemotherapy drugs (carboplatin, etoposide, and paclitaxel) commonly used for lung cancer were selected. Then, the difference in the AUC values between the CKS2-low and the CKS2-high expression groups was evaluated, and the correlation between CKS2 expression level and AUC values was calculated.

### 2.5. Functional Enrichment Analysis

The objects of the Gene Ontology (GO) [18,19] and Kyoto Encyclopedia of Genes and Genomes (KEGG) [20,21] functional enrichment analyses were the differentially expressed genes (DEGs) between the CKS2-low and the CKS2-high expression groups. GO and KEGG enrichment analysis of DEGs through the ‘clusterprofile’ R package were performed to identify the biological functions associated with CKS2. GO enrichment analysis includes three categories: biological processes, molecular function, and cellular component modules. We selected the top 10 significant nodes (*p* < 0.05) for each category in GO enrichment analysis and all significant pathways (*p* < 0.05) in KEGG enrichment analysis to be visualized.

In addition, we also performed Gene Set Enrichment Analysis (GSEA) [22], which is another method to identify functional pathways. This method ranks the genes according to the absolute value of logFC between the CKS2-low and the CKS2-high expression groups and calculates the enrichment score. We used GSEA software for this analysis, and the significance threshold was FDR *q*-value < 0.001.

### 2.6. Statistical Analysis

The comparison between categorical variables was tested using the Chi-square test. Student’s-*t* and ANOVA tests were applied to compare continuous variables between two or more groups if they fit a normal distribution; otherwise, the Wilcoxon Mann–Whitney test was used. Kaplan–Meier (K-M) survival analysis was examined using the log-rank test. The correlation between the two variables was tested using the Spearman method. Two-tailed *p*-values less than 0.05 indicated statistical significance.

## 3. Results

### 3.1. Expression of CKS2 in Stages I–III Invasive Non-Mucinous Lung Adenocarcinoma

By using immunohistochemical staining on 90 pairs of stages I–III invasive non-mucinous lung adenocarcinoma and adjacent normal tissue samples, we found that CKS2 was not only highly expressed in the LUAD samples but also differed significantly between histological subtypes, with the micropapillary subtype having the highest expression (Figure 1A–C). To determine the diagnostic ability of CKS2 to distinguish stages I–III invasive non-mucinous lung adenocarcinoma patients from healthy samples, ROC curves were generated using the TCGA combined with the GTEx dataset (training cohort) and the immunohistochemical scores of 90 samples (validation cohort). The results showed that CKS2 achieved AUC values of 0.939 and 0.849 in the training cohort and in the validation cohort, respectively (Figure 1D–E).

According to the available literature [8], the various tissue subtypes were classified into three categories by their prognostic ability: LEP cluster, PAP/ACI cluster, and SOL/MIP/CRI cluster. The differences in the CKS2 expression among these three clusters were analyzed using the TCGA data and immunohistochemical scores. The results revealed that CKS2 protein and mRNA expression levels were significantly higher in the SOL/MIP/CRI cluster than in the LEP and PAP/ACI clusters (Figure 1F,G).

### 3.2. Prognostic Significance of CKS2 and Different Histological Subtype Clusters

A total of 90 cases of invasive non-mucinous lung adenocarcinoma patients at stages I–III were used in the prognosis analysis. The K-M survival analysis method was adopted for analyzing the differences in overall survival (OS) and recurrence-free progression (RFP) between CKS2-low and CKS2-high expression groups, of which the cutoff value was six scores. Likewise, the differences between various histological subtype clusters were investigated. The results revealed that there were significant differences between CKS2-low and CKS2-high expression groups for both the OS (Figure 2A) and the RFP (Figure 2B), with the OS and the RFP of the CKS2 high-expression group being worse compared with those of the CKS2 low-expression group. Similarly, the OS (Figure 2C) and RFP (Figure 2D) were significantly different in patients with diverse histological subtype clusters, with the SOL/MIP/CRI cluster having the worst OS and RFP. We combined CKS2 with histological subtype clusters for prognostic analysis. According to CKS2 and histological subtype clusters, the samples were sorted into three groups: SOL/MIP/CRI cluster with CKS2 high expression, PAP/ACI cluster with CKS2 low expression, and LEP cluster with CKS2 low expression. Compared to the LEP cluster with low CKS2 expression and the PAP/ACI cluster with low CKS2 expression groups, the SOL/MIP/CRI cluster with high CKS2 expression group exhibited a lower OS and RFP (Figure 2E–H).

### 3.3. Relationship between CKS2 and MKI67 and PCNA

The ‘marker of proliferation Ki-67’ (MKI67) and ‘proliferating cell nuclear antigen’ (PCNA) are clinically important indicators commonly used to assess tumor cell proliferation, as well as the degree of malignancy and prognosis [23,24,25]. To evaluate the association of CKS2 with MKI67 and PCNA, the expression profiles of CKS2, MKI67, and PCNA mRNA (FPKM) in stages I–III invasive non-mucinous lung adenocarcinoma (211 cases) were acquired from TCGA and converted into log2-transformed TPM data. The patients were grouped based on the optimal cut-off value calculated using the R package ‘survminer’. Using the K-M method, MKI67 and PCNA were confirmed as poor prognosis factors in stages I–III invasive non-mucinous lung adenocarcinoma (Figure 2I,J). The result of a correlation test showed that CKS2 was linked positively with MKI67 (Figure 2K) and PCNA (Figure 2L) at the mRNA expression level, indicating that the CKS2 expression level was associated with cell proliferation.

### 3.4. CKS2 Immunohistochemical Score and Clinicopathological Parameters

The relationship between the CKS2 immunohistochemical score and clinicopathological parameters was analyzed to further assess the clinical significance of the CKS2 immunohistochemical score in stages I–III invasive non-mucinous lung adenocarcinoma (Table 1). The CKS2 immunohistochemical score was mainly linked with lymph node metastasis, pathological tumor stage, spread through air spaces (STAS), histological subtypes, and pathological differentiation. The association of CKS2 expression level with both histological subtype and STAS suggested that CKS2 was connected with the growth pattern of invasive non-mucinous lung adenocarcinoma.

### 3.5. CKS2 and Drug Sensitivity

An investigation of the association between CKS2 expression level and the drug sensitivity of LUAD patients to common chemotherapeutic drugs was conducted to explore the application prospects of CKS2 in stages I–III invasive non-mucinous lung adenocarcinoma. Based on log2-transformed TPM data, we obtained the drug sensitivity data of three regularly used chemotherapeutic drugs for lung cancer from the CTRP database (https://portals.broadinstitute.org/ctrp, accessed on 20 December 2021) by ‘oncoPredict’ R package. The AUC value represents the area under the dose–response curve, which represents the degree of drug sensitivity. An increasing AUC value represents a lower drug sensitivity. Analysis of the differences in the sensitivity between CKS2-low and CKS2-high expression groups to different chemotherapeutic drugs indicated that the drug sensitivity of carboplatin and paclitaxel was significantly different in the two groups (Figure 3A–C). A correlation analysis between CKS2 expression and the AUC values revealed that patients with lower expression of CKS2 were more sensitive to carboplatin, while those with higher expression of CKS2 were more sensitive to paclitaxel (Figure 3D–F).

### 3.6. CKS2 and Immune Cell Infiltration 

To investigate the effect of the tumor microenvironment (TME) on CKS2 expression, the CIBERSORT algorithm [26] was applied to identify the relationship between immune cell abundance and CKS2 expression. The algorithm includes 22 common immune cells. According to the results, a variety of immune cells were associated with CKS2 expression, especially T cells (Figure 4A,B).

### 3.7. The Underlying Mechanism of CKS2 Aberrant Expression 

The regulatory mechanism of CKS2 aberrant expression in stages I–III invasive non-mucinous lung adenocarcinoma was explored. Firstly, we used the data from the TCGA database to examine CKS2 gene mutation and copy number variation (CNV) and found that CKS2 gene mutation and CNV were not significantly different between normal lung and tumor tissues.

Then, the effect of DNA methylation and RNA methylation on CKS2 expression was investigated. CpG island methylation data were obtained from TCGA, and RNA methylation level was analyzed using m6A-related genes [27]. The results revealed that between normal lung and tumor tissues, there was no significant difference in the degree of CKS2 CpG island methylation. The correlation between RNA m6A-related genes and CKS2 mRNA abundance was visualized (Figure 4C,D). Six of the thirteen m6A-related genes were not only differentially expressed between the CKS2-low and CKS2-high expression groups but also significantly associated with CKS2 mRNA abundance.

Finally, we analyzed the regulatory function of ceRNAs on CKS2 expression in stages I–III invasive non-mucinous lung adenocarcinoma. CeRNAs are a class of RNA molecule that can bind to miRNA to inhibit its silencing effect on mRNA [28,29]. Using the Wilcoxon Mann–Whitney test, differentially expressed miRNAs and lncRNAs between normal lung and tumor tissues were identified. The differentially expressed miRNAs that interact with CKS2 and the differentially expressed lncRNAs that interact with the differentially expressed miRNAs were obtained from the Starbase database [30]. Correlation analysis identified one differentially expressed miRNA and four differentially expressed lncRNAs which were associated with CKS2 expression. The ceRNA regulatory network related to CKS2 was visualized (Figure 4E). Collectively, CKS2 was regulated not only by methylation modifications but also by ceRNAs in stages I–III invasive non-mucinous lung adenocarcinoma.

### 3.8. Enrichment Analysis of Biological Functions Related to CKS2

To explore the functions of CKS2 in stages I–III invasive non-mucinous lung adenocarcinoma, the patients were grouped according to the median value: CKS2-low and CKS2-high expression groups. Based on the ‘limma’ R package [31], 418 protein-coding DEGs between the two groups were determined with a filter criterion of |log2FC| > 1 and *p*-value < 0.05. 

GO enrichment was conducted on the DEGs (Figure 5A). These genes were primarily concentrated in the biological processes of organelle fission and nuclear division, in the molecular function of tubulin binding and microtubule binding, and in the cellular component of the chromosomal region and spindle formation. KEGG pathway enrichment analysis on the DEGs revealed that these genes were mostly connected to the pathway related to the cell cycle (Figure 5B). We performed GSEA on the CKS2-low and CKS2-high expression groups and found that the CKS2-high expression group had significant differences in genes regulating the cell cycle compared to the CKS2-low expression group (Figure 5C).

### 3.9. CKS2 Protein-Protein Interactions (PPI) Network

From the STRING website [32] we obtained the proteins that interact with CKS2 protein and visualized these interactions using Cytoscape software (Figure 5D). 

## 4. Discussion

Increasing attention is being directed at finding prognostic molecular markers for tumors. The 5-year survival rate for lung cancer, a common malignant tumor, has historically received a lot of attention, yet it is often just about 15% [33]. Even after radical resection, early-stage patients still have a greater risk of recurrence. Therefore, finding effective therapeutic targets or treatment methods is essential for improving the survival rate of patients. The most common subtype of lung adenocarcinoma, which was made up the majority of NSCLC cases, is invasive non-mucinous lung adenocarcinoma.

CKS2 is a cell cycle-related gene that affects cell division. In this study, we found that CKS2 is strongly expressed in stages I–III invasive non-mucinous lung adenocarcinoma and that CKS2 expression in the solid and micropapillary subtypes was significantly higher than that in the Acinar, Papillary, and Lepidic subtypes. Moreover, CKS2 has a good ability in distinguish stages I–III invasive non-mucinous lung adenocarcinoma patients from healthy samples, indicating that CKS2 has the potential to be used as a diagnostic biomarker for stages I–III invasive non-mucinous lung adenocarcinoma. More importantly, high CKS2 expression worsened the prognosis of these LUAD patients. To this end, we analyzed the correlation between CKS2 and MKI67 and PCNA, which were involved in the proliferation of lung adenocarcinoma cells, and found that CKS2 was significantly positively correlated with both of the genes. Furthermore, we found that CKS2 was associated with pathological tumor stage, lymph node metastasis, and the occurrence of STAS in patients. These results suggest that CKS2 may affect the prognosis of patients by affecting the growth pattern of lung adenocarcinoma cells.

Different tumor cells react differently to the same chemotherapeutic drugs, and similar tumor cells have different sensitivities to different chemotherapeutic drugs. Finding the most sensitive chemotherapeutic drugs for diverse tumor patients can not only reduce the adverse effects of chemotherapy drugs on patients but also improve the OS of patients. In this study, it was found that CKS2 expression affected the sensitivity of LUAD cells to carboplatin and paclitaxel. Higher CKS2 expression was associated with higher sensitivity to paclitaxel. In terms of molecular function, CKS2 was found to be linked with microtubule binding by GO enrichment analysis. Paclitaxel is regularly used as a part of adjuvant chemotherapy for lung cancer in clinical treatment. It can promote microtubule polymerization while inhibiting depolymerization, causing the spindle to lose its normal function and cell mitosis to stop [34,35,36]. Therefore, CKS2 may modulate the effect of paclitaxel on LUAD cells by regulating microtubule binding. In contrast, lower CKS2 expression increased the sensitivity of tumor cells to carboplatin. Carboplatin has an important function in antitumor effects by destroying the structure and function of DNA [37]. GO enrichment analysis revealed that CKS2 was correlated with chromosomal regions in terms of cellular component association. From this perspective, CKS2 may affect the function of carboplatin on LUAD cells by influencing carboplatin binding to DNA.

Gene expression is influenced by a multitude of factors, including gene mutation, copy number variation, DNA and RNA methylation, TME, and the regulation of ceRNAs. We found that CKS2 expression level was associated with the T cells in TME, especially activated CD4+ T cells, indicating that the immune microenvironment has an important influence on the expression of CKS2. In addition, the six m6A-related genes associated with CKS2 expression levels include METTL14, WTAP, and ZC3H13 as Writer, HNRNPC as Reader, FTO, and ALKBH5 as Eraser, indicating that the high expression of CKS2 in stages I–III invasive non-mucinous lung adenocarcinoma may be related to the increased RNA stability caused by m6A methylation. As a sponge for adsorbing miRNA, ceRNAs can combine with miRNA to increase the mRNA expression level. In our study, we found that AC026356.1, LINC02535, HELLPAR, and AL024507.2 can act as sponges to adsorb has-miR-30c-5p, which can interact with CKS2, thereby increasing the expression of CKS2.

Through GO, KEGG, and GSEA enrichment analysis, we found that CKS2 was associated with the cell cycle and involved in various biological processes. In addition, the CKS2-related proteins were obtained using the STRING website. Through the UniProt database (https://www.uniprot.org/, accessed on 20 July 2022), we found that the functions of these proteins were connected to the cell cycle. That is, these proteins, together with CKS2, promote tumor growth by regulating the cell cycle of tumor cells.

In clinical work, according to our research, CKS2 expression can be used to evaluate the sensitivity of patients to carboplatin or paclitaxel when considering postoperative adjuvant chemotherapy so as to formulate individualized adjuvant chemotherapy regimens to better improve the overall survival of patients. Additionally, our study revealed that patients with high CKS2 expression were more likely to relapse than those with low CKS2 expression, suggesting that these patients need more attentive and frequent follow-up. In order to perform postoperative adjuvant chemotherapy more efficiently and reduce the risk of postoperative recurrence, clinicians also need to fully consider the characteristics that patients with high CKS2 expression are prone to recurrence.

In contrast to other studies on the expression of CKS2 in LUAD [38,39,40], the primary focus of our research was on invasive lung adenocarcinoma, the most common type of LUAD. Additionally, we investigated in depth the expression of CKS2 in various histological subtypes of invasive lung adenocarcinoma and identified the roles of CKS2 in affecting the drug sensitivity of LUAD tumor cells to carboplatin and paclitaxel. However, the study still had some shortcomings. Firstly, due to the limited sample size collected from a local hospital, the differences in OS between the CKS2-low and CKS2-high expression groups based on different tumor stages are only statistically significant in stage I (Appendix A), and thereby a larger sample cohort is necessary for stratified analysis based on tumor stage. Secondly, the association between CKS2 expression level and the sensitivity to chemotherapeutic drugs has not been verified in clinical practice. Finally, further experimental validations of the predicted CKS2-related functions and mechanisms are necessary.

## 5. Conclusions

In summary, we determined that CKS2 overexpression worsens the prognosis for patients with stages I–III invasive non-mucinous lung adenocarcinoma. For patients with lower CKS2 expression, the outlook is better, and chemotherapy regimens with carboplatin or carboplatin combined with other drugs may be effective. For patients with higher CKS2 expression, chemotherapy regimens with paclitaxel or paclitaxel combined with other drugs can be recommended. CKS2 expression can be useful in determining individualized treatment approaches for LUAD patients.

## Figures and Tables

**Figure 1 cells-11-02611-f001:**
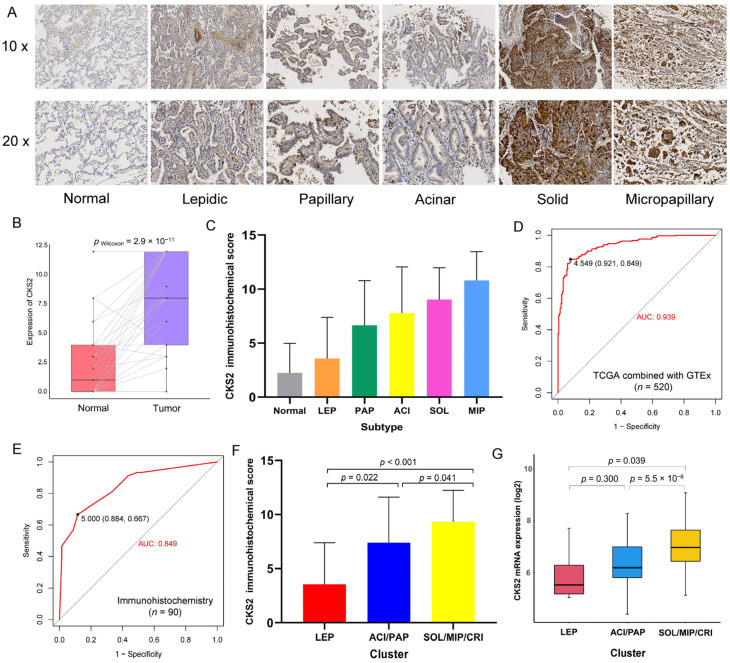
The expression of CKS2 and various histological subtypes in stages I–III invasive non-mucinous lung adenocarcinoma. (**A**) Representative immunohistochemical staining plots for CKS2 in different histological subtypes. (**B**) The difference in CKS2 immunohistochemical scores between 69 pairs of adjacent normal lung and tumor tissues among 90 cases from the local hospital. (**C**) Box plot of the CKS2 immunohistochemical scores (*n* = 90) in different histological subtypes. (**D**) ROC plot of CKS2 mRNA expression predicting LUAD tumorigenesis using TCGA combined with GTEx dataset (training cohort, n = 520, AUC value = 0.939). (**E**) ROC plot of CKS2 immunohistochemical scores predicting LUAD tumorigenesis (validation cohort, *n* = 90, AUC value = 0.849). (**F**) Box plot of CKS2 immunohistochemical scores (*n* = 90) in different histological subtypes clusters. (**G**) Box plot of CKS2 mRNA expression levels (*n* = 211) in different histological subtypes clusters.

**Figure 2 cells-11-02611-f002:**
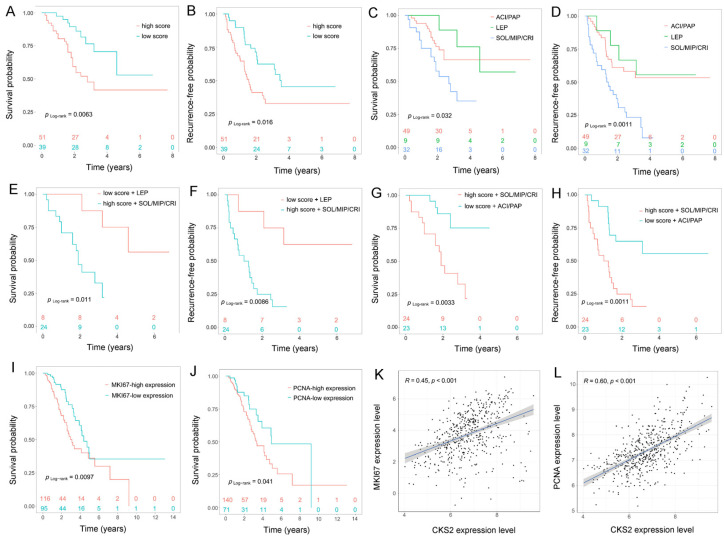
Prognostic significance of CKS2. (**A**,**B**) The difference in OS and RFP between CKS2-low and CKS2-high expression groups. (**C**,**D**) The difference in OS and RFP between various histological subtype clusters. (**E**–**H**) Prognosis analysis of CKS2 combined with different histological subtype clusters. (**I**,**J**) K-M survival analysis of MKI67 and PCNA based on TCGA database. (**K**,**L**) Correlation between CKS2 and MKI67 and PCNA based on TCGA database. OS, overall survival, RFP, recurrence-free progression.

**Figure 3 cells-11-02611-f003:**
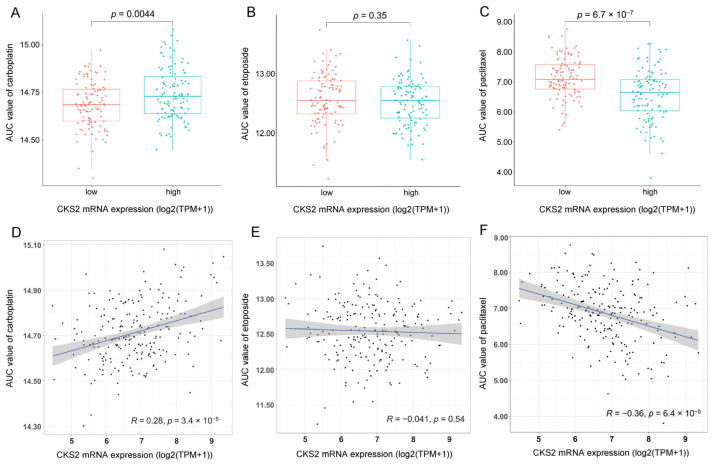
Association between the CKS2 and drug sensitivity. (**A**–**C**) Sensitivity of the CKS2-low and CKS2-high expression groups to different chemotherapeutic drugs. (**D**–**F**) Correlation between CKS2 expression and the AUC value of chemotherapeutic drugs.

**Figure 4 cells-11-02611-f004:**
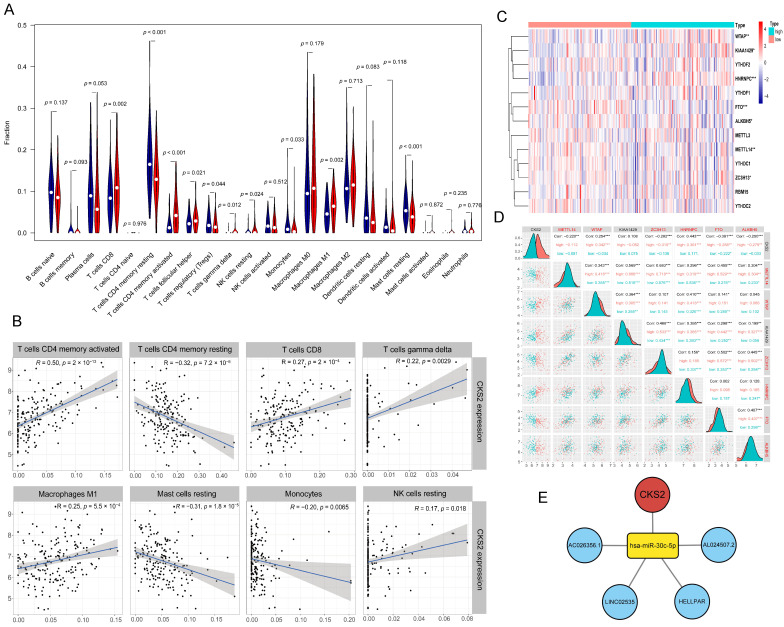
Exploration of the mechanisms for CKS2 overexpression. (**A**) The differences in immune cell abundance between CKS2-low and CKS2-high expression groups. (**B**) Correlation between immune cell abundance and CKS2 expression. (**C**) Heatmap of m6A-related gene expression in CKS2-low and CKS2-high expression groups. (**D**) Relationship between CKS2 expression level and differentially expressed m6a-related genes. (**E**) CeRNA regulatory network of CKS2. * *p* < 0.05, ** *p* < 0.01, *** *p* < 0.001. CeRNA, competing endogenous RNA.

**Figure 5 cells-11-02611-f005:**
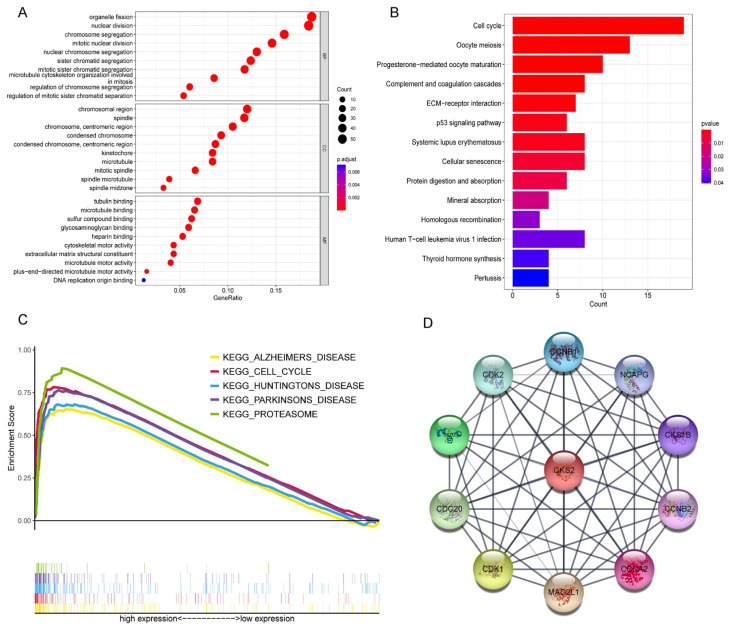
Enrichment analysis related to CKS2. (**A**) GO enrichment for the DEGs between CKS2-low and CKS2-high expression groups. (**B**) KEGG enrichment for these DEGs. (**C**) Gene Set Enrichment Analysis for these DEGs. (**D**) Protein–protein interactions network.

**Table 1 cells-11-02611-t001:** Association between CKS2 immunohistochemical score with clinicopathologic parameter.

Variables	No. of Patients	CKS2 Immunohistochemical Score		*p* Value
		Low Score	High Score	
Total	90	39	51	
Age (year)				0.953
<60	58	25	33	
≥60	32	14	18	
Gender				0.172
male	48	24	24	
female	42	15	27	
Pathological Stage				0.029
Ⅰ–Ⅱ	63	32	31	
Ⅲ	27	7	20	
Pathological T				0.417
T1	56	26	30	
T2	25	9	16	
T3	7	4	3	
T4	2	0	2	
Pathological N				0.024
N0	55	29	26	
N1–N3	35	10	25	
STAS ^1^				0.015
Negative	34	19	15	
Positive	35	9	26	
Histological subtypes				0.013
Solid	27	7	20	
Micropapillary	5	1	4	
Papillary	17	9	8	
Acinar	32	14	18	
Lepidic	9	8	1	
Pathologic differentiation				<0.001
Low	43	8	35	
Median	38	23	15	
High	9	8	1	

^1^ STAS: spread through air spaces. A total of 69 slides met the assessment criteria for STAS, which required sufficient adjacent normal tissue.

## Data Availability

Transcriptome data and partial pathological slide images involved in this article can be obtained from TCGA and GEO database.

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
