# Peer review of "Cyclin-Dependent Kinase Subunit 2 (CKS2) as a Prognostic Marker for Stages I–III Invasive Non-Mucinous Lung Adenocarcinoma and Its Role in Affecting Drug Sensitivity"

_cells, 2022, doi:10.3390/cells11162611_

Round 1
Reviewer 1 Report
Major revision required:
1/ Section 2. Materials and Methods; 2.1. Clinicopathological slides: please provide the detailed information about the treatment of the 90 stage I-III LUAD patients. This is very important because the Authors perform the survival analyses and any differences in treatment may affect the results.
2/ Section 2. Materials and Methods; 2.3. Immunohistochemical staining: please, provide the exact clone name of each antibody used in IHC.
2/ Section 3. Results, 3.1. Expression of CKS2 in stages I-III invasive non-mucinous lung adenocarcinoma, Figure 1: Please, provide more detailed information in the figure legend which graph/data set corresponds to IHC or mRNA expression. Explain whether the data concerns only TCGA cases (211 cases) or all the cases included into the study (211 cases from TCGA + 90 cases from Hospital of Zhengzhou University). Provide the number of each group/subgroup in the figure legend (n= X). Each figure and legend should be self-explanatory. Please, apply this comment to each figure.
3/ I recommend to call TCGA group as, for example, the 'training cohort' and the cases from Hospital of Zhengzhou University as a 'validation cohort'. Currently, it is difficult to recognize which group is evaluated in a given section.
4/ Section 3. Results, 3.2. Prognostic significance of CKS2 and different histological subtype clusters: Why survival analyses were performed for 90 LUAD patients presenting stages I-III? In NSCLC, overall survival depends greatly on the stage of the disease at diagnosis. The Authors should perform more detailed analysis of survival for each subgroup of patients presenting different stage.
5/ Section 4. Discussion: Please, discuss more thorougly what is clinical usefulness of the study results. The Authors showed that CKS2 expression affected the sensitivity of LUAD 293 cells to carboplatin and paclitaxel and that there were significant differences between CKS2-low and CKS2-high expression groups for both the OS and the RFP. However, in patients with I-III stage LUAD usually the most effective treatment strategy is tumor resection and the expression of certain genes may change due to the disease course. What is the practical use of the prognostic data from primary tumors for the patients after disease recurrence?
6/ Section 4. Discussion: the discussion is too brief. Please, address all the results gained in the study and refer to other studies evaluating CKS2 expression in LUAD, e.g. PMID: 33083148, PMID: 34729099, PMID: 34333494.
Reviewer 2 Report
In this manuscript, the authors showed that CKS2 can serve as a prognostic biomarker for non-mucinous lung adenocarcinoma and modulate the effect of paclitaxel and carboplatin. This result extends our understanding of the CKS2, in addition, the results of CKS2 and drug sensitivity are novel. Moreover, several points as indicated below need to be addressed by authors to improve the quality of the article.
1. In Figure 5D, authors obtained the proteins that interact with CKS2 proteins. Can you explain how these proteins interact for cancer growth?
2. The resolution of figures is low.
Round 2
Reviewer 1 Report
Major revision:
1. In their response to Reviewers, the Authors presented the results of more detailed analysis of survival for each subgroup of patients presenting different stage (including K-M plots). I suggest that these data should be included in the Supplementary material to the article and referenced in the paragraph on the limitations of the study.
2. In the Discussion section, the Authors did not referred to other studies evaluating CKS2 expression in LUAD, e.g. PMID: 33083148, PMID: 34729099, PMID: 34333494. Since there are data from former studies, it is important to demonstrate what new knowledge has been brought by the present study in this subject. I suggest to refer to other studies evaluating CKS2 expression in LUAD in the Discussion (and add new references).
